# A logistic regression model based on inpatient health records to predict drug-induced liver injury caused by ramipril—An angiotensin-converting enzyme inhibitor

**Phuong Nguyen Thi Thu**[1,2]\*, **Mai Ngo Thi Quynh**[1], **Hung Nguyen Van**[1], **Hoi Nguyen Thanh**[2], **Khue Pham Minh**[1]

**1** Hai Phong University of Medicine and Pharmacy, Hai Phong, Vietnam, **2** Hai Phong International Hospital, Hai Phong, Vietnam

\* nttphuong@hpmu.edu.vn

## Abstract

Drug-induced liver injury (DILI) is a rare side effect of angiotensin-converting enzyme inhibitors (ACEIs). Ramipril is a widely used ACE compound because of its effectiveness in the treatment of hypertension and heart failure, as well as its low risk of adverse effects. However, the clinical features of ramipril, and the risk of DILI, have not been adequately studied. A retrospective cohort study was performed based on data from 3909 inpatients to compare the risk of DILI conferred by ramipril and other ACEIs. A logistic regression model was then constructed and validated against data from 1686 patients using ramipril, of which 117 patients were diagnosed with DILI. The use of ramipril increased the risk of DILI by 2.68 times (odds ratio = 2.68; 95% confident interval (CI):1.96–3.71) compared with the group using other ACEIs. The clinical features of DILI in the ramipril group were similar to those from the ACEI group (P>0.05), except that the ALT level was higher (P<0.05). A logistic regression model including Body mass index (BMI), comorbidity, liver disease, daily dose, alanine aminotransferase (ALT), and alkaline phosphatase (ALP) was built and successfully validated for DILI risk prediction, with the area under the receiver operating characteristic curve of the model of 0.82 (95% CI: 0.752–0.888). We recommend that clinicians should be aware of the levels of ALT and ALP as well as BMI, comorbidities, and liver disease before prescribing ramipril to avoid the risk of DILI in patients.

## Introduction

Drug-induced liver injury (DILI) can occur after multiple exposures to prescription and over-the-counter medicines through a wide range of mechanisms [1]. The global prevalence of DILI was predicted annually ranging from 10 to 15 per 10,000 to 100,000 persons used to prescription drugs [2, 3]. Numerous risk factors have been identified to be linked to the development of DILI. Overall, age has been associated with a higher risk of DILI (with the notable exception of DILI for valproic acid, which frequently occurs in children). Hepatocellular injury is more

**Data Availability Statement:** All relevant data are within the article and its Supporting information files.

**Funding:** This work is funded by Haiphong University of Medicine and Pharmacy.

**Competing interests:** The authors have declared that no competing interests exist.

prevalent in the younger age group than cholestatic and mixed liver injury. Female sex appears to be more susceptible to DILI than male sex. One of the explanations for the higher risk of DILI seen in females is their smaller body size [4, 5]. More than 1000 drugs and herbal supplements involved in the appearance of DILI, and the numbers continue to grow [6]. The U.S. National Institutes of Health maintains a database of drugs, herbal medicines, and dietary supplements related to the development of DILI. Among them, ramipril is an angiotensin-converting enzyme (ACE) inhibitor used in the treatment of hypertension and heart failure. Ramipril is hydrolyzed by carboxylesterase 1 enzyme in the liver to its active carboxylic metabolite ramiprilat, but undergoes little further liver metabolism. Ramipril is a well-tolerated drug and has been mainly related to mild adverse effects, such as cough [7]. Common adverse drug reactions of ramipril have been reported including dizziness, fatigue, headache, cough, digestive disorders, and skin rash. Ramipril is associated with a low rate of transient serum aminotransferase increases and has been linked to rare cases of DILI [8]. However, only a few cases of ramipril resulting in liver injury have been documented, but the rare instances that have been published have resembled typical ACE inhibitor-related acute liver injury [9, 10]. Previous studies have demonstrated the hepatotoxic potential of ramipril, highlighting the need for physicians to be vigilant about this problem [10].

Vietnam is a developing country and is a member of the Association of Southeast Asian Nations, with a population of approximately 91.7 million. Although the National Center for Drug Information and Adverse Drug Reactions monitoring (the DI & ADR center) was established in 2009 to collect and assess ADR reports nationwide, the low ADR reporting rate among healthcare professionals was documented, including educational training and lack of time [11, 12]. Although ramipril is an extremely popular ACE inhibitor in Vietnam, it has not been reported for ADRs according to the national pharmacovigilance system. This may be because Vietnamese health professionals often focus on severe or fatal ADRs.

We first conducted a retrospective study to assess the risk factors of DILI with ramipril and establish a logistic regression model to estimate the risk of DILI with ramipril in inpatients among Vietnamese people, which can improve the clinician's ability to identify patients at a risk of developing DILI during ramipril use.

## Materials and methods

Our study was conducted in 2021 and included data collected from the electronic medical records of inpatients from 2018 to 2020 at Hai Phong International General Hospital.

### Subjects

The study was divided into two phases:

1. To assess the risk of DILI with ramipril compared to that with ACEI compounds
   Inclusion criteria:

- Patients aged $\geq$18 years

- Patients have liver function tests, including alanine aminotransferase (ALT) and alkaline phosphatase (ALP) tests before and during ACEI treatment.

- Patients using ACEIs with ALT $<3 \times$ upper limit of normal (ULN) and ALP $<2 \times$ ULN on admission. The ULN of ALT was 40 U/L, while that of ALP was 140 U/L.

2. To build a model to predict the risk of causing DILI with ramipril, we selected the following patients
   Inclusion criteria:

- Patients aged ≥18 years using ramipril.

- Patients have complete ALT and ALP tests before and during ramipril treatment.

- A hospital stay duration of more than 48h for the follow-up of altered values, allowing for the confirmation of possible DILI.

DILI was detected using the Roussel Uclaf Causality Assessment Method scales. This method includes time to onset, course of ALT after taking drugs, alcohol use, age, concomitant drugs, previous drug hepatotoxicity, and response to unintentional drug re-exposure. For each criterion, the widest scale ranged from -3 to +3 with 7 degrees (-3, -2, -1, 0, +1, +2, and +3), corresponding to the increasing probability of the role of the evaluated drug. The total score was classified into 5 degrees: ≤ 0, excluded; 1–2, unlikely; 3–5, possible; 6–8, probable; and ≥ 9, highly probable [13].

The follow-up period of the study was 1 year from the date of DILI detection. For instance, patients diagnosed with DILI in 2019 were followed up until 2019. Similarly, those who were first diagnosed with DILI in 2019 were followed up until 2020.

**Drug study.**   All patients in ramipril group were administered ramipril at a dose of 2.5 mg or 5 mg. The other ACEIs that the patients took were enalapril (2.5 mg/5 mg/10 mg/20 mg), perindopril (5 mg), lisinopril (5 mg/10 mg/20 mg), and captopril (12,5 mg/25 mg/50 mg).

## Data processing

First, we selected 5279 in patients whom ACEIs were indicated. A total of 480 patients were excluded due to a lack of liver function testing before or during drug administration. A total of 4799 ACEI patients underwent screening for ALT, ALP, DILI causality assessments and clinical diagnosis. Subsequently, 1061 patients had results suspicious for DILI while 3732 were excluded for having no features of DILI. In the group with suspected DILI, we excluded 890 patients because other drugs were suspected to have caused DILI. Ultimately, we selected 177 patients with DILI based on laboratory tests and physicians' diagnoses on medical records, including 117 patients using ramipril and 60 patients using other ACEIs. The patient selection process is illustrated in Fig 1.

Data were collected from two groups. The data from group 1 included basic information such as age, sex, weight, height, BMI, date of admission, number of days hospitalized,

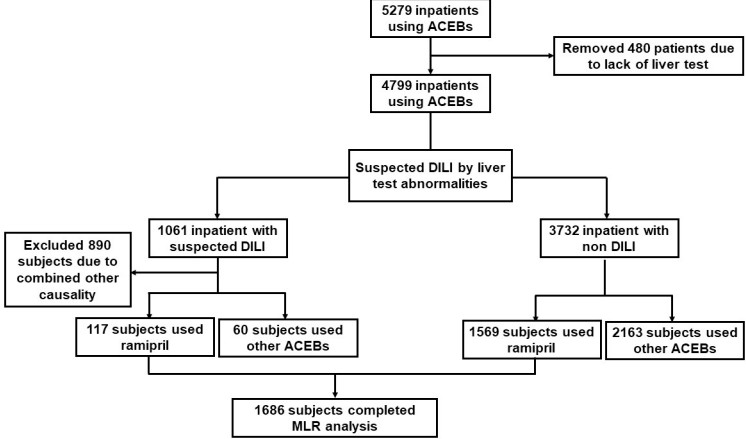

**Fig 1. Data processing.** ACEIs: angiotensin converting enzyme inhibitors; DILI: Drug-induced liver injury; MLR: multiple logistic regression.

diagnosis, and history of liver disease, while those from group 2 included information related to DILI, such as liver test before and after prescription, drugs, and dose.

## Statistical analysis

**Model development.** Fig 2 presents a scheme illustrating the variables used to build the model. The logistic regression model was built using the R statistical software, version 3.2.4 (A Language and Environment for Statistical Computing, Vienna, Austria) [14]. To develop the DILI risk estimation model, we mapped the variables collected by the MHR to 23 discrete variables. The criteria for the inclusion of a variable in our model were based on clinical and statistical significance. We used forward selection based on the chi-square test of the change in residual deviance. Our goal was to build a logistic regression model that minimizes the variables until the optimal model that describes the data is found. We constructed three risk estimation models. These models used the presence or absence of DILI as the dependent variable and the 23 aforementioned variables as independent variables. We used a cutoff value of P<0.001 to add new terms. Differences among groups were analyzed using the chi-square test for qualitative variables, one-way analysis of variance for continuous variables with normal distributions, and the non-parametric Kruskal–Wallis test for continuous variables with non-standard distributions.

**Model evaluation.** We used a 2-fold cross-validation technique to evaluate the predictive performance of the three models. In general, we randomized the data into two sets, a1 and a2, to ensure that both sets were equal in size. We then trained our models on the a1 set and validated on the a2 set, followed by training on the a2 and validating on the a1. This methodology avoided the issue of validating the model using the same data used to estimate the parameters by using separate estimation and evaluation subsets of the data. Specifically, we divided the dataset into two subsets to ensure that all findings were independent of each other. We started with the first fold to estimate the coefficients of the independent variables (training data) and predicted the probability of DILI on the 2nd fold (test data). It is of note that for the inclusion of variables in the final model, we used the whole dataset (1686 records), which gave us the best possible estimates of the variables from the available data.

Logistic regression is considered to provide a better fit to the data if it demonstrates an improvement over a model with fewer predictors. Predictor variables were removed from the

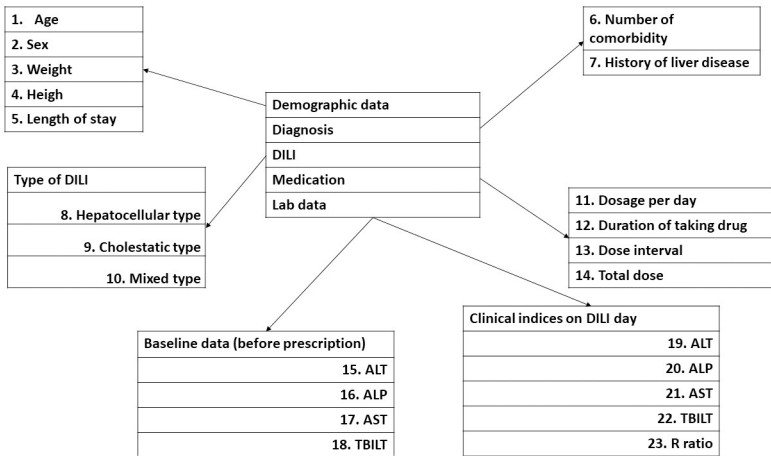

**Fig 2. Variables star schema.** DILI, drug induced liver injury; ALT, alanine aminotransferase; AST, aspartate aminotransferase; ALP, alkaline phosphatase; TBIL, total bilirubin.

first model to the second and third models. Given that the null hypothesis stated that the reduced model was true, a p-value for the overall model fit statistic of <0.05 would warrant a rejection of the null hypothesis.

**Performance measures.** We measured the performance of the three models using the outcome (probability of DILI) of the test set obtained by 2-fold cross-validation. We plotted and measured the area under the receiver operating characteristics (ROC) curve of Model-1, Model-2, and Model-3 using the probability of DILI for comparing three areas under ROC curves obtained from the same dataset [15]. Brier score was used to check the goodness of a predicted probability score [16]. Lower Brier scores exhibited improved precision [17]. The Youden index method was used to determine the optimal cut-off point, which is the difference between the number of true positives and the number of false positives over all possible cut-off points.

## Ethical considerations

The study protocols were reviewed and approved by the Institutional Review Board (IRB) of Hai Phong International Hospital, Vietnam (IRB. 20.318). The study was conducted in accordance with the Declaration of Helsinki and the International Conference on the Harmonization of the Technical Requirements for the Registration of Pharmaceuticals for Human Use—Good Clinical Practice guidelines. Prior to data collection, ethical approval was obtained from the ethics subcommittees of Hai Phong International Hospital. All subjects provided written informed consent prior to the commencement of the study.

## Results

Table 1 shows the clinical and laboratory characteristics of study participants.

Table 2 shows DILI in patients using ACEIs and the corresponding odds ratio (ORs) for DILI. The use of ramipril was strongly related to the risk of DILI compared to the use of other ACEIs. The OR for DILI among patients taking ramipril was 2.68 (1.96–3.71), with a P-value <0.001 (Table 2).

Table 3 shows the clinical and laboratory characteristics of patients using ACEIs with DILI. We divided this group into two subgroups consisting of the group that used ramipril (n = 117) and the group that used different ACEIs (n = 60). Patients with suspected DILI due to ramipril had an average age of 64.2 years, and 57% of patients were men. This characteristic was similar to the group of patients with suspected DILI from ACEIs with p-values of 0.54 and 0.44,

**Table 1. Characteristics of study participants (n = 1686).**

| Clinical data | Value |
|---|---|
| Age, years (mean ± SD) | 66.48±10.04 |
| Weight, kg (mean ± SD) | 45±12.4 |
| Male (n (%)) | 999 (59.2) |
| BMI (mean ± SD) | 22.41±4.29 |
| Length of stay, days (mean ± SD) | 14 ±4.5 |
| Duration of taking drug, days (mean ± SD) | 30.1± 8.7 |
| Laboratory characteristics (mean ± SD) | |
| ALT, U/l | 37.9±67.1 |
| AST, U/l | 37.5±92.4 |
| ALP, U/l | 37.1±70.7 |
| TBiL, µmol/l | 1.3±5.5 |

**Table 2. The relationship between taking angiotensin-converting enzyme inhibitors and ORs for drug-induced liver injury (n = 3909).**

| Treatment | Non-DILI (n (%)) | DILI (n (%)) | OR (95%Cl) | P-value |
|---|---|---|---|---|
| Other ACEIs | 2163 (58.0) | 60 (33.9) | 1.0 | |
| Ramipril | 1569 (42.0) | 117 (66.1) | 2.68 (1.96–3.71) | <0.001 |
| Total (n) | 3732 (100) | 177 (100) | | |

DILI, drug-induced liver injury; ACEIs, angiotensin-converting enzyme inhibitor; OR, odds ratio; CI, confidence interval.

respectively. The group of patients stayed in the hospital for a long period, with an average time of 24 days in both groups. The duration of ramipril use in the ramipril group was similar to that of other ACEIs. On the day of prescription medication causing DILI, the ALT test results of the ramipril group revealed an average value of 31.8 ± 74.6 UI/L, which was similar to that in the group treated with other ACEIs (16.1± 18.1 UI/L; p = 0.11). Similar to the ALP results, the group that used ramipril with an average of 35.9±104.4, this result was not different from the group without ramipril with p = 0.24. However, on the day DILI was diagnosed, the ramipril group had a 14.1-fold increase in ALP from 35.9±104.4 (UI/L) to 191.5±236.2. In the group using other ACEIs, this rate increased by approximately 11.3 times from 24.9±20.4 (UI/

**Table 3. Comparison of clinical characteristics of drug-induced liver injury patients taking ramipril (n = 117) or other angiotensin-converting enzyme inhibitors (n = 60).**

| Clinical data | Ramipril group (n = 117) | Other ACEIs group (n = 60) | P-value |
|---|---|---|---|
| Age, years (mean ± SD) | 64.2±10.4 | 62.7±11.4 | 0.40 |
| Male (n (%)) | 67 (57) | 38 (63) | 0.54 |
| BMI (mean ± SD) | 24.4±4.4 | 25.6±4.2 | 0.09 |
| Length of stay, days (mean ± SD) | 24±3.8 | 24±5.2 | 0.44 |
| Time to onset of DILI, days (range) | 31 (13–250) | 31 (15–245) | 0.24 |
| Duration of taking drug, days (mean ± SD) | 29.8±5.9 | 31.9±10.6 | 0.18 |
| Clinical indices on prescription day (mean ± SD) | | | |
| ALT, U/l | 31.8±74.6 | 16.1±18.1 | 0.11 |
| AST, U/l | 26±39.6 | 44.3±48.9 | 0.15 |
| ALP, U/l | 35.9±104.4 | 24.9±20.4 | 0.24 |
| TBiL, μmol/l | 5±15.7 | 5.4±6.6 | |
| Clinical indices on DILI day (mean ± SD) | | | |
| ALT, U/l | 245.7±292.6 | 166.9±123.5 | 0.03 |
| AST, U/l | 137±93.4 | 91±14.6 | 0.14 |
| ALP, U/l | 191.5±236.2 | 159.1±123.5 | 0.17 |
| TBiL, μmol/l | 7.1±15.7 | 5.5±13.6 | 0.51 |
| Clinical types of DILI | | | |
| Hepatocellular type (n, %) | 1 (1) | 0 | 1 |
| Cholestatic type (n, %) | 107 (91) | 55 (92) | 1 |
| Mixed type (n, %) | 9 (8) | 5 (8) | 1 |

DILI, drug-induced liver injury; ACEI, angiotensin-converting enzyme inhibitor; ALT, alanine aminotransferase; AST, aspartate aminotransferase; ALP, alkaline phosphatase; TbiL, total bilirubin; BMI, body mass index; SD, standard deviation.

**Table 4. Model 1: Multiple logistic-regression results relating the factors of drug-induced liver injury.**

| Coefficients | Estimate | Odds ratio (95% CI) | P-value |
|---|---|---|---|
| Intercept | -13.32 | 0.01(0.01,0.01) | 0.001 |
| Age (years) | -0.03 | 0.98(0.93,1.03) | 0.342 |
| Sex (Male) | 0.39 | 1.47(0.52,4.26) | 0.473 |
| BMI | 0.2 | 1.21(1,1.47) | 0.051 |
| Number of comorbidities (n) | 0.39 | 1.48(1.07,2.05) | 0.018 |
| Liver disease | 2.08 | 7.96(1.64,34.06) | 0.007 |
| Total dose in 24 hours (mg) | 0.37 | 1.45(1.2,1.81) | 0.001 |
| Total bilirubin | 0.01 | 1.01(0.91,1.11) | 0.959 |
| Baseline ALT | 0.04 | 1.04(1.02,1.06) | 0.001 |
| Baseline ALP | 0.05 | 1.05(1.03,1.07) | 0.001 |

BMI, Body mass index; CI, confidence interval; ALT, alanine aminotransferase; ALP, alkaline phosphatase;

L) to 159.1± 123.5. However, this difference was not statistically significant (p = 0.25). Regarding the ALT test, on the date of determination of DILI in the ramipril group, this level was increased 8.2 times, the proportion in the group using other ACEIs was 7.9 (p = 0.78).

Most patients diagnosed with DILI in our study were classified as cholestatic type mechanisms, with the incidence in the ramipril group of 107 (91%) and in the other group of ACEIs, 55 (92%) (P = 1). Very few patients were classified as mixed type, with only 9 patients in the ramipril group and 5 patients in the other ACEI group. Out of 177 patients using ACEIs with DILI in our study, only one patient was classified as hepatocellular type. In our study, apart from ramipril, patients were indicated with other ACEIs including enalapril, imidapril, lisinopril, and perindopril, with proportions of 2.16%, 0.07%, 32.78%, and 65%, respectively.

In Model-1, nine independent variables (mammographic features and demographic factors) were found to be significant in predicting DILI (Table 4). The most important predictors associated with DILI, as identified by this model, were BMI, the number of comorbidities, history of liver disease, total dose of ramipril in 24 h, and the level of ALT and ALP pre-prescription. Age was not found to be a significant predictor, but it was included in the model because of its clinical relevance. In Model-2, which included seven independent variables which were significant in predicting the risk of DILI (Table 5), including age, BMI, the number of comorbidities, history of liver disease, total dose of ramipril in 24 h, and the level of ALT and ALP pre-prescription. In Model-3, which included only six independent variables which were significant in predicting the risk of DILI (Table 6), including BMI, the number of comorbidities,

**Table 5. Model 2: Multiple logistic-regression results relating the factors of drug-induced liver injury.**

| Coefficients | Estimate | Odds ratio (95% CI) | P-value |
|---|---|---|---|
| Intercept | -12.55 | 0.01(0.01,0.01) | 0.001 |
| Age | -0.03 | 0.98(0.93,1.03) | 0.298 |
| BMI | 0.18 | 1.19(0.99,1.43) | 0.063 |
| Number of comorbidities (n) | 0.4 | 1.48(1.08,2.05) | 0.017 |
| Liver disease | 2.02 | 7.5(1.55,31.41) | 0.008 |
| Total dose in 24 hours (mg) | 0.36 | 1.43(1.19,1.78) | 0.001 |
| Baseline ALT | 0.04 | 1.04(1.02,1.06) | 0.001 |
| Baseline ALP | 0.05 | 1.05(1.03,1.07) | 0.001 |

BMI, Body mass index; CI, confidence interval; ALT, alanine aminotransferase; ALP, alkaline phosphatase;

**Table 6. Model 3: Multiple logistic-regression results relating the factors of drug-induced liver injury.**

| Coefficients | Estimate | Odds ratio (95% CI) | P-value |
|---|---|---|---|
| Intercept | -14.73 | 0.01(0.01,0.01) | 0.001 |
| BMI | 0.2 | 1.22(1.02,1.46) | 0.03 |
| Number of comorbidities (n) | 0.37 | 1.44(1.05,1.98) | 0.024 |
| Liver disease | 2.17 | 8.73(1.86,35.56) | 0.004 |
| Total dose in 24 hours (mg) | 0.37 | 1.45(1.22,1.79) | 0.001 |
| Baseline ALT | 0.04 | 1.04(1.02,1.06) | 0.001 |
| Baseline ALP | 0.05 | 1.05(1.04,1.07) | 0.001 |

BMI, Body mass index; CI, confidence interval; ALT, alanine aminotransferase; ALP, alkaline phosphatase;

history of liver disease, the total dose of ramipril in 24 h, and the level of ALT and ALP pre-prescription. We assessed the significance of the variables in all three models (as shown in Tables 4–6) using the test dataset.

As shown in Table 4, the application of comorbidities increased the risk of DILI by 1.48-fold (OR = 1.48, p = 0.018). In comparison to standard hypertension patients, the risk of DILI in patients who were diagnosed with liver diseases such as cirrhosis, hepatitis, and fatty liver disease increased the risk of DILI by 7.96-fold (OR = 7.96, p = 0.007). The risk of DILI increased by 45% for each additional 1 mg dose of ramipril (OR = 1.45, p = 0.001). The risk of DILI increased by 4% and 5% for 1UI/L higher levels of ALT and ALP before ramipril pre-scription, respectively. However, age and sex had no effect on the risk of DILI (OR = 0.98, p = 0.342), (OR = 1.47, p = 0.474), respectively.

For model 2 (Table 5, Fig 3), the application of comorbidities increased the risk of DILI by 1.48-fold (OR = 105, p = 0.017). In comparison with normal hypertension patients, the risk of DILI in patients who were diagnosed with liver diseases increased the risk of DILI by 7.5-fold

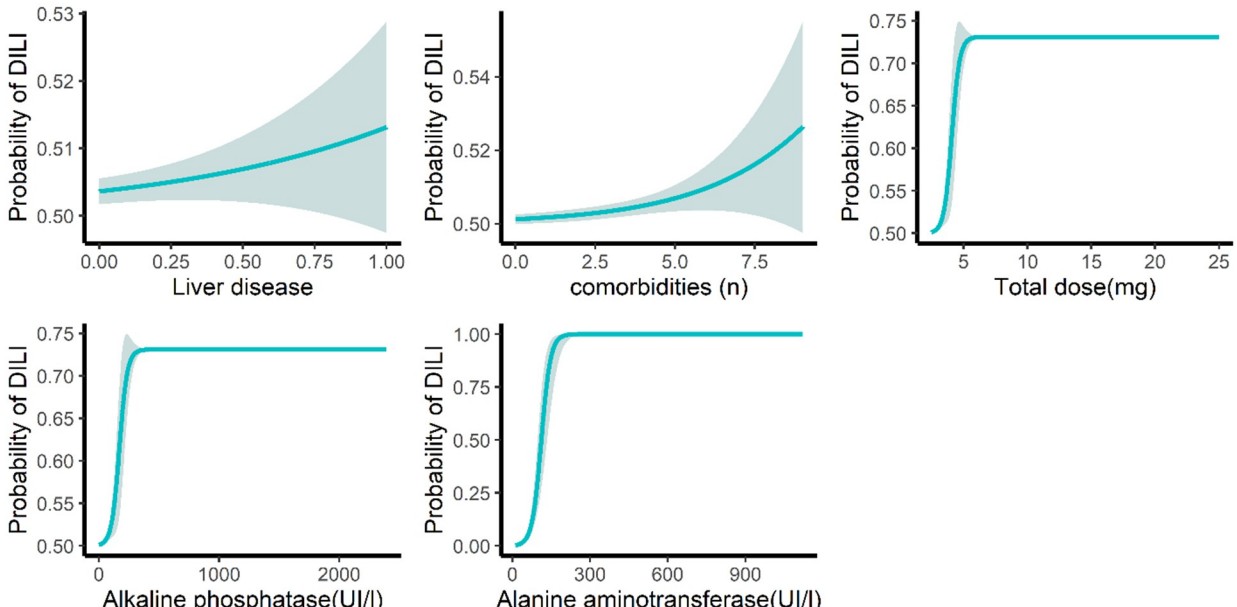

**Fig 3. Impact of independent predictors on the probability of drug induced liver injury on multiple logistic-regressions.** DILI: drug induced liver injury.

**Table 7. Comparison of the performances and the evaluationsof the models.**

| Model | Cut-off point | Specificity | Sensitivity | P-value# | $A_{ROC}$! | Brier scrore | 95% CI$ | AIC | $R^2$ |
|---|---|---|---|---|---|---|---|---|---|
| Model 1 | 0.029 | 0.884 | 0.727 | | 0.824 | 0.039 | 0.766–0.897 | 147.17 | 0.362 |
| Model 2 (- sex, BILT) | 0.024 | 0.865 | 0.727 | 0.32 | 0.83 | 0.04 | 0.758–0.892 | 143.69 | 0.370 |
| Model 3 (-age, sex, BILT) | 0.033 | 0.896 | 0.682 | 0.29 | 0.827 | 0.028 | 0.752–0.888 | 142.8 | 0.366 |

$A_{ROC}$: Area under the receiver operating characteristic curve; AIC: Akaike information criterion

(OR = 7.5, p = 0.008), and the risk of DILI increased by 43% for each additional 1 mg dose of ramipril (OR = 1.43, p = 0.001). The risk of DILI increased by 4% and 5% for 1UI/L higher levels of ALT and ALP before ramipril prescription, respectively.

For model 3 (Table 6), compared to patients with no liver disease, the risk of DILI in patients who were diagnosed with liver disease, the risk of DILI by 8.73-fold (OR = 8.73, p = 0.004); the risk of DILI increased by 45% for each additional 1 mg dose of ramipril (OR = 1.45, p = 0.001). The risk of DILI increased by 4% and 5% (p = 0.001) for 1UI/L higher levels of ALT and ALP before ramipril prescription, respectively.

With the cutoff point of 0.029 Model 1 achieved an $A_{ROC}$ of 0.831 (0.766–0.897), as measured by the DILI assessment assigned to each record. Model 2 achieved an $A_{ROC}$ of 0.825 (0.758–0.892), which was not significantly different (P = 0.32) than Model 1's $A_{ROC}$ (Table 7, Fig 4). Model 3 achieved an $A_{ROC}$ of 0.825 (0.758–0.892), which was not significantly different (P = 0.29) from that of Model 1. The results show that model 3 did indeed provide a significantly better fit to the data compared to models 1 and 2 (Table 7, Fig 4).

We found that model 3 was better with the smallest Brier score and showed that $A_{ROC}$ was not significantly different from model 1 and model 2 (p>0.05). Furthermore, model 3 also released a sensitivity and specificity of 0.682 and 0.896, respectively.

## Discussion

From our study, we found that the odds of developing DILI are approximately two and a half times higher with ramipril use compared to without, and a multivariate logistic regression model was developed and validated using data from 1686 inpatients using ramipril.

Clinically apparent liver injury due to ACE inhibitors is rare. The different ACE inhibitors were documented separately as case reports. There have been over fifty reports of drug-

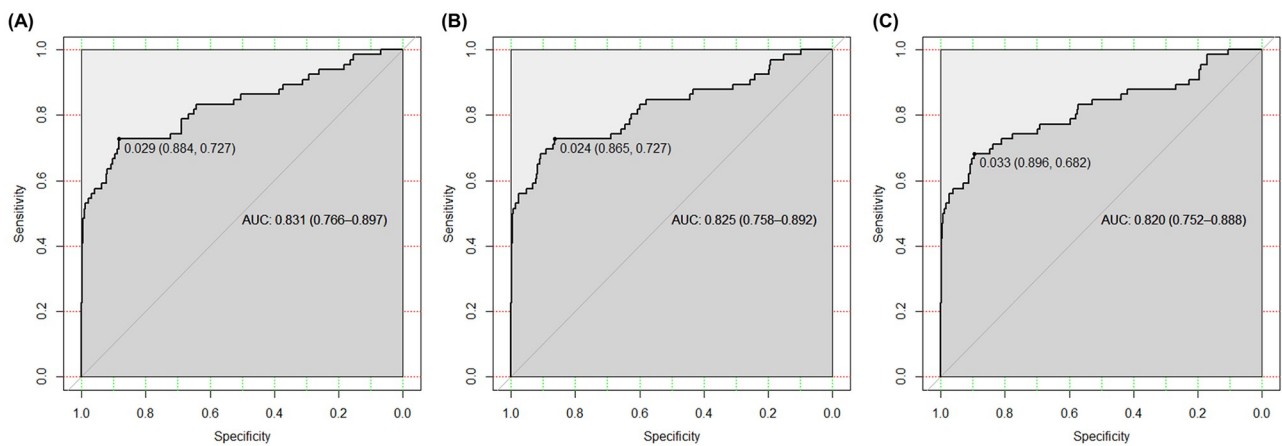

**Fig 4. Comparison of receiver operating characteristic curve curves of 3 models.** AUC: area under the receiver operating characteristic curve.

induced hepatotoxicity associated with other ACEIs, the most frequently being captopril [18]. These studies confirmed that hepatotoxicity is a well-established, yet rare, adverse effect of captopril. Following captopril, enalapril has been reported to be most associated with liver injury [10, 19, 20]. For ramipril, there are five case reports of ramipril-induced DILI [9, 10, 21]. Taking ramipril group was observed to be higher risk of developing DILI compared to other ACEIs. Captopril was not prescribed in our hospital, which led to a lack of comprehensive comparison. However, our patients used enalapril, imidapril, lisinopril, and perindopril with the same proportion as ramipril use. Cholestasis appears to be the most common liver injury pattern in ACEI-induced hepatotoxicity, and this pattern has been evident in two previous cases of ramipril-induced liver injury, with biopsies revealing cholestasis and bile duct necrosis [9]. We also found that cholestasis was the most frequent mechanism of DILI in both groups using ramipril and non-ramipril, with 91% and 92%, respectively. Our study demonstrated that the time to onset of DILI caused by either ramipril or other ACEIs was 31 days, ranging from 13 to 240 days in the ramipril group and from 15 to 245 days in the other ACEI groups. This finding is similar to that of other studies showing that adverse events from ACEIs can occur from 10 days to 8 months [10, 22]. In addition, increased ALT and ALP levels was association with DILI in patients who received ramipril. Ramipril was shown to cause an increase of 20-fold in ALT and a 5-fold increase in ALP levels in a 40-year-old male patient after 10 months using ramipril [10]. Two other patients reported an increase in ALP levels to 957 UI/l and 507 UI/l, while ALT levels did not [21]. Our study indicated that the level of ALT on the day of onset of ramipril was higher than that on the day of onset of other ACEIs.

Our results showed that model 3 of six independent predictors including bmi, comorbidity, total dose of ramipril, baseline ALT, baseline ALP, and liver disease performed better than the other models with $A_{ROC}$ = 082(0.752–0.888). In addition, the specificity of model 3 were better than those of models 1 and 2 for predicting DILI non-recovery, and binary logistic regression analysis was used to construct a number of variables, including digestive symptoms, jaundice, TBIL, and DBIL [23]. In our case, TBIL had no significant impact on DILI risk prediction. Therefore, we did not include these terms in the logistic regression model. To assess the severity of DILI risk associated with oral medications, a DILI score model based on the daily dose, lipophilicity, and formation of reactive metabolites was built [24]. Indeed, our study also indicated that the daily dose of ramipril is a strong predictor of DILI risk assessment (OR = 1.45, p = 0.001). Ramipril also possesses properties that can increase the risk of DILI higher than other compounds, such as an active metabolite and lipophilic compared to enalapril, which is hydrophilic ACEI [25, 26].

A multiple logistic regression model identified variables independently associated with death within 6 months of presenting with suspected DILI, including comorbidities, model for end-stage liver disease score, and serum level of albumin at presentation [27]. Similarly, our model found that comorbidities were a strong burden of DILI. We also considered liver disease in our multiple variables model, which increased the risk of liver injury by 3.77-fold.

The aim of our model is to aid decision-making by generating a risk prediction at the prescription time point. At the time of prescribing, clinicians need a simple and accurate model to predict the risk of DILI for patients, we chose a model with only five independent variables. For this reason, we excluded the clinical characteristics of patients on the DILI day, such as symptoms, liver test results, and recovery.

This case-control study tested a hypothesis about the correlation between intake of ramipril and DILI. Admittedly, our study is not as powerful as the other studies with a different study design in terms of establishing a causal relationship. Nevertheless, our study provided data and a reference for future clinical studies that would be using more rigorous scientific methods. Another limitation of our study is the patient's adherence to medication because our study

subjects are outpatients. However, to measure adherence to drug therapy, we used the patient's self-report in every hospital visit. Moreover, identification of the cause of DILI may be influenced by certain drugs or supplements that cause DILI, but to the best of our knowledge, this has not been reported in the literature. Although the independent variables were statistically significant related to DILI appreance our model performance had a low R-squared value. The $R^2$ value is an overall measure of strength of association between the model and the response and does not reflect the contribution of independent predictor variable.

In conclusion, our study indicated that the use of ramipril can increase the risk of liver injury to a greater extent than other ACEI compounds. Furthermore, we developed and validated our logistic regression models (Model-3) that can effectively predict the risk of DILI. We recommend that clinicians should be aware of ALT, ALP, bmi, comorbidities, and liver disease before prescribing ramipril to avoid the risk of DILI in patients. Our study supports the idea that further prospective studies are needed to confirm the effect of our model in routine clinical practice to improve overall performance.

## Supporting information

**S1 Data.**
(XLSX)

## Acknowledgments

We thank all the staff members for their contributions to this study.

## Author Contributions

**Conceptualization:** Phuong Nguyen Thi Thu.

**Data curation:** Phuong Nguyen Thi Thu, Hung Nguyen Van, Hoi Nguyen Thanh, Khue Pham Minh.

**Formal analysis:** Phuong Nguyen Thi Thu, Mai Ngo Thi Quynh, Hoi Nguyen Thanh.

**Investigation:** Phuong Nguyen Thi Thu, Hoi Nguyen Thanh, Khue Pham Minh.

**Methodology:** Phuong Nguyen Thi Thu, Hung Nguyen Van.

**Project administration:** Phuong Nguyen Thi Thu, Hung Nguyen Van, Khue Pham Minh.

**Software:** Mai Ngo Thi Quynh.

**Validation:** Mai Ngo Thi Quynh, Khue Pham Minh.

**Visualization:** Phuong Nguyen Thi Thu, Mai Ngo Thi Quynh.

**Writing – original draft:** Phuong Nguyen Thi Thu, Khue Pham Minh.

**Writing – review & editing:** Phuong Nguyen Thi Thu, Mai Ngo Thi Quynh, Khue Pham Minh.

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
