## [Decision Letter · Decision Letter 0]

15 Jul 2021

PONE-D-21-13763

A logistic regression model based on inpatient health records to predict drug-induced liver injury caused by ramipril – an angiotensin-converting enzyme inhibitor

PLOS ONE

Dear Dr. Nguyen,

Thank you for submitting your manuscript to PLOS ONE. After careful consideration, we feel that it has merit but does not fully meet PLOS ONE’s publication criteria as it currently stands. Therefore, we invite you to submit a revised version of the manuscript that addresses the points raised during the review process.

We look forward to receiving your revised manuscript.

Kind regards,

Edford Sinkala

Academic Editor

PLOS ONE

Additional Editor Comments:

There is need to include limitations in this study. Discussion section has a lot of numbers that are supposed to be confined to the result section. Please discuss the results in the discussion section

Journal Requirements:

In addition, please include the date of ethical approval.

Reviewers' comments:

Reviewer's Responses to Questions

**Comments to the Author**

1. Is the manuscript technically sound, and do the data support the conclusions?

Reviewer #1: Partly

Reviewer #2: Yes

2. Has the statistical analysis been performed appropriately and rigorously? 

Reviewer #1: No

Reviewer #2: Yes

3. Have the authors made all data underlying the findings in their manuscript fully available?

Reviewer #1: Yes

Reviewer #2: Yes

4. Is the manuscript presented in an intelligible fashion and written in standard English?

Reviewer #1: Yes

Reviewer #2: No

5. Review Comments to the Author

Reviewer #1: Reviewer comments

i. In line 25 in the abstract there is an error

ii. In line 40, there is a statement that “age has been associated with a higher risk of Drug Induced Liver Injury” but it is not clear the direction of association.

iii.

iv. In line 41, the authors have indicated that “female sex appear to be more susceptible to DILI compared to man” Authors should consider using a more appropriate adjective “male”

v. In line 44, the authors indicated the National Institutes of Health maintains a searchable database of drug but there is no indication where this institute is. This should be made clear.

vi. The authors assessed the risk factors of DILI during ramipril and establish a logistic regression model to estimate the DILI of inpatients using ramipril in Vietnamese people. How the authors did insured that all the participants in this study were Vietnamese since they were using retrospective hospital records? What was the definition of someone to be Vietnamese in this study?

vii. The authors have indicated that the used data for the year 2018 to 2020. What was the justification for using data between this periods? What was the defined follow-up time in this study since it is a retrospective cohort study?

viii. In this study there is no indication when it was conducted.

ix. In line 73, authors have indicated that this study was reviewed and approved by the Hai Phong International Hospital Institutional Review Board, Vietnam but there is no reference to provide evidence.

x. In line 74, there is a statement that this study was conducted in accordance with the Declaration of Helsinki and the International Conference on the Harmonization of the Technical Requirements for the Registration of Pharmaceuticals for Human Use - Good Clinical Practice guidelines. This deserves to be referenced.

xi. Even if this data was obtained from some database, include epidemiological questionnaire that was also used to gather sociodemographic and clinical data for this study. Please explain thoroughly how the questionnaire was designed, implemented, filled, and checked for accuracy and completeness.

xii. In line 78, what ACEB compounds?

xiii. In line 82, there is an indication that ALT and ALP had upper normal limit. What were these limits?

xiv. Authors need to indicate the duration of exposure to ramipril and dosage as well as those on ACEIs.

xv. Since this study used clinical assessment as part of DILI diagnosis and it is possible that different physicians with different clinical experience could have led to similar patients classified differently or different patients classified in the same group resulting in misclassification. How did authors dealt with this issue?

xvi. Specifically, what clinical parameters were used for DILI diagnosis by clinicians?

xvii. It is important for authors to report the other drugs patients were on apart from ramipril since this not a randomized clinical trial. Further, dosages and duration of treatment for ramipril and ACEIs that were used should be reported.

xviii. In line 100, the authors have indicated that they collected data on height, weight and BMI. BMI can be important in explain DILI but this variable and others has not be reported anywhere in the results section. What could be the possible explanation for this omission?

xix. In line 101 one of the variable collected was history but it is not clear. History about what?

xx. In line 106 authors have indicated that they mapped the variables collected by the MHR to 32 discrete variables and they have referred to Figure 2 which is not detailed. It is not clear what these 32 variables were?

xxi. In line 111, R statistical software should include manufacturer, name and country should be added in the manuscript.

xxii. Line 111, authors used forward variable selection method when constructing logistic regression model. What was the justification for choosing this method as oppose to other methods? Further, in forward selection inclusion of a new variable may make an existing variable in the model non-significant. How did the authors dealt with this drawback? If so how?

xxiii. In 128, the use of abbreviation H0 should first be written in full before using it even if it is a well- known abbreviation.

xxiv. Line 138, indicates that ethical approval was obtained from the ethics subcommittees of the Hai Phong International Hospital. Authors need to provide evidence by including the reference number.

xxv. In this study, authors have claimed that rimipril caused DILI. It is important to recognize that causality cannot be established definitively through a retrospective cohort study which is an observation study; however, such studies are powerful tool that can provide important evidence to suggest causality, as well as information regarding the strength of an association between ramipril and DILI. What authors have reported is an association between ramipril and DILI.

xxvi. It is important for authors to give a biological plausibility to establish the observed relationship and identify a scientifically valid mechanism by which the ramipril can associated with DILI.

xxvii. Surprisingly, authors have not included limitations in this study and the findings of this study could be distorted by many factors not limited to biases, misclassification and measurement error, confounding and effect modification (also called “interaction”).

xxviii. In line 147, under the results for table 1, the footnotes are not complete.

xxix. In line 151, authors have reported that “average age of 64.2, and the majority were men (57%)”. Authors should explain the central measure of tendency used with units and why? Further, the reported percentage should be accompanied by an absolute number.

xxx. In line 154, there is a statement that “on the day of admission, the ALT test results of the ramipril group averaged 31.8 74.6 (UI 155 / L)” This should be made clearer to the readership. This applies to a number of sections where such results have been reported.

xxxi. In a number of sections such lines 158 and 159, authors have not indicated the reported central measure of tendency. This should be made clear.

xxxii. In table 2, authors have reported (n±SD) suggesting number and standard deviation for age and length of hospital stay. This is not appropriate. Standard deviation should be reported with appropriate measure of central tendency. This applies to the other tables in the results section.

xxxiii. What statistical tests were performed to obtain the p-values for different variables in table 2 and other tables in the results?

xxxiv. Although there are some footnotes under table 2, there are some still missing. This applies to table 3a.

xxxv. What were the comorbidity in the study?

xxxvi. The authors used multivariate logistic regression model. Was this really multivariate logistic regression or multivariable regression?

xxxvii. In the discussion the authors keep repeating the estimate (odds ratio) and p-value. This is repetition of results. Authors should just discuss!

xxxviii. In line 245, there is a statement that “on the day of DILI, the ALT and ALP levels in our patient were elevated by 8.2” Which particular patient is being referred to and why?

xxxix. Authors should make it clear whether which method of analysis they used. Either intention to treat analogue or per protocol analysis and why?

xl. Authors have recommended that clinicians should be aware of age and liver disease before prescribing ramipril to avoid the risk of DILI in patients. This is surprising since age showed no difference between DILI and ACEIs groups as well as liver disease at 97.5% CI. What was the rational for this recommendation?

Reviewer #2: A well written paper on an important subject. The following corrections should be made.

Line 65, the word first is out of place, one expects it to be followed by a second action, which is not done in this sentence.

Line 72-26 - Subjects

This sentence belongs to the ethics section, not where it is currently.

Line 154-156 - the ± sign is missing from the numbers in the text and needs to be corrected.

Analysis

Sex and duration/how long someone has been on treatment, although found to be important factors in other studies, have not been included in the model or discussed at all. This needs to be explained.

Discussion

A discussion of factors associated with DILI from Ramipril (from the modeling) needs to be placed within the context of findings from similar studies in the literature. This has not been done. Are your findings novel, similar or different from those found by similar studies?

Limitations

Authors need to discuss the limitations of their findings. Why should we believe these findings?

6. PLOS authors have the option to publish the peer review history of their article (what does this mean?). If published, this will include your full peer review and any attached files.

Reviewer #1: **Yes: **Dr Patrick Kaonga, University of Zambia, School of Public Health, Department of Epidemiology and Biostatistics

Reviewer #2: No

---

## [Author Response · Author response to Decision Letter 0]

29 Jul 2021

Editor comments

Thank you for your comment. We have prepared our files according to the PLOS ONE’s requirements

2. Please provide additional details regarding participant consent. 

Thank you for your suggestion. The Ethical considerations was revised as below:

The study protocols were reviewed and approved by the Institutional Review Board (IRB) of Hai Phong International Hospital, Vietnam (IRB. 20.318). The study was conducted in accordance with the Declaration of Helsinki and the International Conference on the Harmonization of the Technical Requirements for the Registration of Pharmaceuticals for Human Use - Good Clinical Practice guidelines. Prior to data collection, ethical approval was obtained from the ethics subcommittees of Hai Phong International Hospital. All subjects provided written informed consent prior to the commencement of the study. 

3. In your Data Availability statement, you have not specified where the minimal data set underlying the results described in your manuscript can be found. PLOS defines a study's minimal data set as the underlying data used to reach the conclusions drawn in the manuscript and any additional data required to replicate the reported study findings in their entirety. All PLOS journals require that the minimal data set be made fully available.

Thank you for your comment.

 We uploaded our study’s minimal underlying data set as a Supporting Information file

Reviewer #1: Reviewer comments

i. In line 25 in the abstract there is an error

Thank you for your comment. We revised our manuscript based on your suggestion.

ii. In line 40, there is a statement that “age has been associated with a higher risk of Drug Induced Liver Injury” but it is not clear the direction of association.

Thank you for your comment. We have edited and clarified the relation between age and DILI in the introduction

iv. In line 41, the authors have indicated that “female sex appear to be more susceptible to DILI compared to man” Authors should consider using a more appropriate adjective “male”

Thank you for your comment. We revised our manuscript based on your suggestion.

v. In line 44, the authors indicated the National Institutes of Health maintains a searchable database of drug but there is no indication where this institute is. This should be made clear.

Thank you for your comment. We revised our manuscript based on your suggestion.

vi. The authors assessed the risk factors of DILI during ramipril and establish a logistic regression model to estimate the DILI of inpatients using ramipril in Vietnamese people. How the authors did insured that all the participants in this study were Vietnamese since they were using retrospective hospital records? What was the definition of someone to be Vietnamese in this study?

Thanks for your question. At the patient information system, clinicians (doctors, nurses) collected information about the patient's race. Thanks to that, we can easily identify the population of the research subjects who are Vietnamese

vii. The authors have indicated that the used data for the year 2018 to 2020. What was the justification for using data between this periods? What was the defined follow-up time in this study since it is a retrospective cohort study?

Thank you for your question. 

The follow-up period of the study was 1 year from the date of DILI detection. For instance, patients diagnosed with DILI in 2019 were followed up until 2019. Similarly, those who were first diagnosed with DILI in 2019 were followed up until 2020.

This explanation has been added to the manuscript

viii. In this study there is no indication when it was conducted.

Thank you for your suggestion. We revised our manuscript as below:

Our study conducted in 2021 including data collected from the electronic medical records of inpatients from 2018 to 2020 at Hai Phong International General Hospital. 

ix. In line 73, authors have indicated that this study was reviewed and approved by the Hai Phong International Hospital Institutional Review Board, Vietnam but there is no reference to provide evidence.

Thank you for your suggestion. Research review process by the hospital's IRB committee according to the procedures of the Ministry of Health of Vietnam. The approval of the Board is in official writing, however it is written in Vietnamese. We are ready to provide if the Review Board and the Editorial Board request

x. In line 74, there is a statement that this study was conducted in accordance with the Declaration of Helsinki and the International Conference on the Harmonization of the Technical Requirements for the Registration of Pharmaceuticals for Human Use - Good Clinical Practice guidelines. This deserves to be referenced.

Thank you for your suggestion. The Ethical considerations was revised as below:

The study protocols were reviewed and approved by the Hai Phong International Hospital Institutional Review Board, Vietnam. The study was conducted in accordance with the Declaration of Helsinki and the International Conference on the Harmonization of the Technical Requirements for the Registration of Pharmaceuticals for Human Use - Good Clinical Practice guidelines. Prior to data collection, ethical approval was obtained from the ethics subcommittees of the Hai Phong International Hospital. All subjects gave written informed consent before study initiation. If the subject is a child, a consent letter to participate in the study was obtained from the guardian 

xi. Even if this data was obtained from some database, include epidemiological questionnaire that was also used to gather sociodemographic and clinical data for this study. Please explain thoroughly how the questionnaire was designed, implemented, filled, and checked for accuracy and completeness.

Thank you for your comment. RUCAM is the main tool for us to evaluate the causality if a DILI is suspect. This is a scale that has been developed, evaluated and widely used in the US, Europe and the world since 1989 (1) (2). We added the follow paragraph on Methods:

DILI was detected according to the Roussel Uclaf Causality Assessment Method (RUCAM) scales containing time to onset, course of ALT after taking drugs, alcohol use, age, concomitant drugs, previous drug hepatotoxicity, and response to unintentional drug re-exposure. For each criterion, the widest scale ranged from -3 to +3 with 7 degrees (-3, -2, -1, 0, +1, +2, +3), corresponding to the increasing probability of the role of the evaluated drug. The total score was classified into 5 degrees: ≤ 0, excluded; 1–2, unlikely; 3–5, possible; 6–8, probable; and ≥9, highly probable (3)

xii. In line 78, what ACEB compounds?

Thank you for your comments. We added the required data as below:

All patients in ramipril group were administered ramipril at a dose of 2.5 mg or 5 mg. The other ACEIs that the patients took were enalapril (2.5 mg/5 mg/10 mg/20 mg), perindopril (5 mg), lisinopril (5 mg/10 mg/20 mg), and captopril (12,5 mg/25 mg/50 mg).

xiii. In line 82, there is an indication that ALT and ALP had upper normal limit. What were these limits?

Thank you for your comments. We added information in Methods as below:

The ULN of ALT was 40 U/L, while that of ALP was 140 U/L.

xiv. Authors need to indicate the duration of exposure to ramipril and dosage as well as those on ACEIs.

Thank you for your comment. We added the data as your suggestion into Table 2. The data interpretation was added in the results

The duration of ramipril use in the ramipril group was similar to that of other ACEIs

xv. Since this study used clinical assessment as part of DILI diagnosis and it is possible that different physicians with different clinical experience could have led to similar patients classified differently or different patients classified in the same group resulting in misclassification. How did authors dealt with this issue?

Thanks for your comment. Clinicians diagnose DILI according to the RUCAM scale. This is supplemented and explained in the methods section. This scale helps different doctors to diagnose DILI equally.

xvi. Specifically, what clinical parameters were used for DILI diagnosis by clinicians?

DILI was detected according to the Roussel Uclaf Causality Assessment Method (RUCAM) scales containing time to onset, course of ALT after taking drugs, alcohol use, age, concomitant drugs, previous drug hepatotoxicity, and response to unintentional drug re-exposure. For each criterion, the widest scale ranged from -3 to +3 with 7 degrees (-3, -2, -1, 0, +1, +2, +3), corresponding to the increasing probability of the role of the evaluated drug. The total score was classified into 5 degrees: ≤ 0, excluded; 1–2, unlikely; 3–5, possible; 6–8, probable; and ≥9, highly probable (3)

xvii. It is important for authors to report the other drugs patients were on apart from ramipril since this not a randomized clinical trial. Further, dosages and duration of treatment for ramipril and ACEIs that were used should be reported.

Thank you for your comment. We added the drug study section that described more detail as follow:

Drug study:

All patients in ramipril group were administered ramipril at a dose of 2.5 mg or 5 mg. The other ACEIs that the patients took were enalapril (2.5 mg/5 mg/10 mg/20 mg), perindopril (5 mg), lisinopril (5 mg/10 mg/20 mg), and captopril (12,5 mg/25 mg/50 mg).

xviii. In line 100, the authors have indicated that they collected data on height, weight and BMI. BMI can be important in explain DILI but this variable and others has not be reported anywhere in the results section. What could be the possible explanation for this omission?

BMIs of the two groups were similar and did not contribute statistically to the development of the constructed model. Therefore, we did not include BMI in the model building variables. However, to be more explicit about the clinical indicators we collected, we have added BMI data for both groups to Table 2.

xix. In line 101 one of the variable collected was history but it is not clear. History about what?

Thank you for your comment. We have revised our manuscript as request

xx. In line 106 authors have indicated that they mapped the variables collected by the MHR to 32 discrete variables and they have referred to Figure 2 which is not detailed. It is not clear what these 32 variables were?

Thank you for your comment. We made the mistake of forgetting to cite figure 2. I added figure 2 to account for the variables evaluated during modeling.

xxi. In line 111, R statistical software should include manufacturer, name and country should be added in the manuscript.

Thank you for your comment. We have revised our manuscript as request

xxii. Line 111, authors used forward variable selection method when constructing logistic regression model. What was the justification for choosing this method as oppose to other methods? Further, in forward selection inclusion of a new variable may make an existing variable in the model non-significant. How did the authors dealt with this drawback? If so how?

Thank you for your comment. Logistic regression is a classification algorithm used to find the probability of event success and event failure. It is used when the dependent variable is binary(0/1, True/False, Yes/No) in nature. It is suitable for the purpose of developing our model to predict DILI occurrence using ramipril.

In our study, we deal with negative confounders by using multivariate analysis. Multivariate models can handle large numbers of covariates (and also confounders) simultaneously. For example in our study that aimed to measure the relation between 23 variables in the same model. Logistic regression is considered to provide a better fit to the data if it demonstrates an improvement over a model with fewer predictors.

xxiii. In 128, the use of abbreviation H0 should first be written in full before using it even if it is a well- known abbreviation.

Thank you for your comment. We have revised our manuscript as request

xxiv. Line 138, indicates that ethical approval was obtained from the ethics subcommittees of the Hai Phong International Hospital. Authors need to provide evidence by including the reference number.

Thank you for your comment. We have revised our manuscript as request

xxv. In this study, authors have claimed that rimipril caused DILI. It is important to recognize that causality cannot be established definitively through a retrospective cohort study which is an observation study; however, such studies are powerful tool that can provide important evidence to suggest causality, as well as information regarding the strength of an association between ramipril and DILI. What authors have reported is an association between ramipril and DILI.

We totally agree with your opinion. Our study provided for the first time an association between ramipril use and DILI. Other studies described only individual cases. We tried to reduced the bias of drug exposure. The identification of the cause of DILI was determined not only by the patient's memory but also by the carefully maintained medical records.. The limitations of the study were also added to the discussion with the following content:

This case-control study tested a hypothesis about the correlation between intake of ramipril and DILI. Admittedly, our study is not as powerful as the other studies with a different study design in terms of establishing a causal relationship. Nevertheless, our study provided data and a reference for future clinical studies that would be using more rigorous scientific methods. Moreover, identification of the cause of DILI may be influenced by certain drugs or supplements that cause DILI, but to the best of our knowledge, this has not been reported in the literature. 

xxvi. It is important for authors to give a biological plausibility to establish the observed relationship and identify a scientifically valid mechanism by which the ramipril can associated with DILI.

Thank you for your comments. We showed the scientifically mechanism and evidence from literature about relationship of ramipril and DILI in introduction and discussion sections as below:

However, only a few cases of ramipril resulting in liver injury have been documented, but the rare instances that have been published have resembled typical ACE inhibitor-related acute liver injury (9, 10). Previous studies have demonstrated the hepatotoxic potential of ramipril, highlighting the need for physicians to be vigilant about this problem (10). 

Until now there are five case reports of ramipril-induced DILI (9, 10, 18).

xxvii. Surprisingly, authors have not included limitations in this study and the findings of this study could be distorted by many factors not limited to biases, misclassification and measurement error, confounding and effect modification (also called “interaction”).

The limitations of the study were also added to the discussion with the following content:

This case-control study tested a hypothesis about the correlation between intake of ramipril and DILI. Admittedly, our study is not as powerful as the other studies with a different study design in terms of establishing a causal relationship. Nevertheless, our study provided data and a reference for future clinical studies that would be using more rigorous scientific methods. Moreover, identification of the cause of DILI may be influenced by certain drugs or supplements that cause DILI, but to the best of our knowledge, this has not been reported in the literature. 

xxviii. In line 147, under the results for table 1, the footnotes are not complete.

Thank you for your comment. We have revised our manuscript as request

xxix. In line 151, authors have reported that “average age of 64.2, and the majority were men (57%)”. Authors should explain the central measure of tendency used with units and why? Further, the reported percentage should be accompanied by an absolute number.

Thank you for your comment. We have revised our manuscript as request

xxx. In line 154, there is a statement that “on the day of admission, the ALT test results of the ramipril group averaged 31.8 74.6 (UI 155 / L)” This should be made clearer to the readership. This applies to a number of sections where such results have been reported.

Thank you for your comment. We have revised our manuscript as below:

On the day of prescription medication causing DILI, the ALT test results of the ramipril group averaged 31.8 ±74.6 (UI / L), similar to the group using other ACEIs 16.1± 18.1 with p=0.11.

xxxi. In a number of sections such lines 158 and 159, authors have not indicated the reported central measure of tendency. This should be made clear.

Thank you for your comment. We have revised our manuscript

xxxii. In table 2, authors have reported (n±SD) suggesting number and standard deviation for age and length of hospital stay. This is not appropriate. Standard deviation should be reported with appropriate measure of central tendency. This applies to the other tables in the results section.

Thank you for your comment. We have revised with mean as central tendency.

xxxiii. What statistical tests were performed to obtain the p-values for different variables in table 2 and other tables in the results?

Thank you for your comment. We added required information in statistic analysis section

xxxiv. Although there are some footnotes under table 2, there are some still missing. This applies to table 3a.

Thank you for your comment. We have revised our manuscript

xxxv. What were the comorbidity in the study?

Thank you for your comment. In our study, we did not describe any type of comorbidities. We only counted the number of comorbidities diagnosed according to the ICD-10 disease code

xxxvi. The authors used multivariate logistic regression model. Was this really multivariate logistic regression or multivariable regression?

Thank you for question. We used the method of logistic regression for developing a classification algorithm used to find the probability of DILI.

xxxvii. In the discussion the authors keep repeating the estimate (odds ratio) and p-value. This is repetition of results. Authors should just discuss!

Thank you for your comment. We have revised our manuscript as your suggestion.

xxxviii. In line 245, there is a statement that “on the day of DILI, the ALT and ALP levels in our patient were elevated by 8.2” Which particular patient is being referred to and why?

Thank you for your comment. We have revised our manuscript as below:

On the day of DILI, the ALT and ALP levels in patients using ramipril were elevated by 8.2-and 14.1-fold, respectively

xxxix. Authors should make it clear whether which method of analysis they used. Either intention to treat analogue or per protocol analysis and why?

Thank you for your question. To make the method of statistic analyis clearer, we revised this section as below:

The logistic regression model was built using the R statistical software, version 3.2.4 (A Language and Environment for Statistical Computing, Vienna, Austria (14). We used forward selection based on the chi-square test of the change in residual deviance. Differences among groups were analyzed using the chi-square test for qualitative variables, one-way analysis of variance for continuous variables with normal distributions, and the non-parametric Kruskal‐Wallis test for continuous variables with non-standard distributions. We used a cutoff value of P<0.001 to add new terms.

xl. Authors have recommended that clinicians should be aware of age and liver disease before prescribing ramipril to avoid the risk of DILI in patients. This is surprising since age showed no difference between DILI and ACEIs groups as well as liver disease at 97.5% CI. What was the rational for this recommendation?

Thank you for this review. In Table 1, we only compared the median ages in the ramipril group and other comparable ACEIs, but we did not compare liver disease. However, liver disease was a significant risk factor when we compared groups with and without DILI in the ramipril-using population (Tables 3a,b,c). When building a logistic linear regression model, we find the prevalence of DILI in the group of subjects using ramipril (1687 with non_DILI and 117 subjects with DILI) (Figure 1)

Reviewer #2: A well written paper on an important subject. The following corrections should be made.

Line 65, the word first is out of place, one expects it to be followed by a second action, which is not done in this sentence.

Thank you for your comment. We have revised our manuscript

Line 72-26 - Subjects

This sentence belongs to the ethics section, not where it is currently.

Thank you for your comment. We have revised our manuscript as your suggestion

Line 154-156 - the ± sign is missing from the numbers in the text and needs to be corrected.

Thank you for your comment. We have revised our manuscript as your suggestion

Analysis

Sex and duration/how long someone has been on treatment, although found to be important factors in other studies, have not been included in the model or discussed at all. This needs to be explained.

Thank you for your comment. Gender characteristics and duration of drug use have been added and described in Table 2 and discussed in the Discussion section

Discussion

A discussion of factors associated with DILI from Ramipril (from the modeling) needs to be placed within the context of findings from similar studies in the literature. This has not been done. Are your findings novel, similar or different from those found by similar studies?

Our study provided for the first time an association between ramipril use and DILI. Other studies described only individual cases. This described in introduction and discussion sections as below:

Thank you for your comment

However, only a few cases of ramipril resulting in liver injury have been documented, but the rare instances that have been published have resembled typical ACE inhibitor-related acute liver injury (9, 10). Previous studies have demonstrated the hepatotoxic potential of ramipril, highlighting the need for physicians to be vigilant about this problem (10). 

Until now there are five case reports of ramipril-induced DILI (9, 10, 18).

Limitations

Authors need to discuss the limitations of their findings. Why should we believe these findings?

Thank you for your comment

The limitations of the study were also added to the discussion with the following content:

This case-control study tested a hypothesis about the correlation between intake of ramipril and DILI. Admittedly, our study is not as powerful as the other studies with a different study design in terms of establishing a causal relationship. Nevertheless, our study provided data and a reference for future clinical studies that would be using more rigorous scientific methods. Moreover, identification of the cause of DILI may be influenced by certain drugs or supplements that cause DILI, but to the best of our knowledge, this has not been reported in the literature. 

REFERENCE

1. Danan G, Teschke R. Drug-induced liver injury: why is the Roussel Uclaf Causality Assessment Method (RUCAM) still used 25 years after its launch? Drug safety. 2018;41(8):735-43.

2. Teschke R, Wolff A. Regulatory causality evaluation methods applied in kava hepatotoxicity: are they appropriate? Regulatory Toxicology and Pharmacology. 2011;59(1):1-7.

3. Danan G, Benichou C. Causality assessment of adverse reactions to drugs--I. A novel method based on the conclusions of international consensus meetings: application to drug-induced liver injuries. J Clin Epidemiol. 1993;46(11):1323-30.

---

## [Decision Letter · Decision Letter 1]

9 May 2022

PONE-D-21-13763R1A logistic regression model based on inpatient health records to predict drug-induced liver injury caused by ramipril – an angiotensin-converting enzyme inhibitorPLOS ONE

Dear Dr. Nguyen,

Thank you for submitting your manuscript to PLOS ONE. After careful consideration, we feel that it has merit but does not fully meet PLOS ONE’s publication criteria as it currently stands. Therefore, we invite you to submit a revised version of the manuscript that addresses the points raised during the review process. PLOS ONE's publication criteria require that the data presented in PLOS ONE manuscripts must support the conclusions drawn (http://journals.plos.org/plosone/s/criteria-for-publication#loc-4). We have some concerns about the conclusions in the manuscript. This is a retrospective observational study. However, here there are claims related to causation, e.g. "Our study demonstrated that using ramipril increases the risk of DILI" and "We found that taking ramipril increased the 2.68-fold risk of DILI (OR=2.68, p<0.001) compared to other ACEIs". Please revise your manuscript to address any instances of claims that are not supported by your results, e.g lines 251-252, lines 259-260 and lines 266-268 and also address the limitations of your study.

We look forward to receiving your revised manuscript.

Kind regards,

Miquel Vall-llosera Camps

Senior Editor

PLOS ONE

Journal Requirements:

Reviewers' comments:

Reviewer's Responses to Questions

**Comments to the Author**

1. If the authors have adequately addressed your comments raised in a previous round of review and you feel that this manuscript is now acceptable for publication, you may indicate that here to bypass the “Comments to the Author” section, enter your conflict of interest statement in the “Confidential to Editor” section, and submit your "Accept" recommendation.

Reviewer #1: All comments have been addressed

2. Is the manuscript technically sound, and do the data support the conclusions?

Reviewer #1: Yes

3. Has the statistical analysis been performed appropriately and rigorously? 

Reviewer #1: Yes

4. Have the authors made all data underlying the findings in their manuscript fully available?

Reviewer #1: Yes

5. Is the manuscript presented in an intelligible fashion and written in standard English?

Reviewer #1: Yes

6. Review Comments to the Author

Reviewer #1: All comments that were raised in the previous version of the manuscript have been addressed. In my opinion the manuscript is ready to be published in PLOS ONE.

7. PLOS authors have the option to publish the peer review history of their article (what does this mean?). If published, this will include your full peer review and any attached files.

Reviewer #1: **Yes: **Patrick Kaonga - Post-doc, PhD, MPH

---

## [Author Response · Author response to Decision Letter 1]

10 May 2022

Dear editor and reviewers,

Thank you for giving us the opportunity to submit a revised draft of the manuscript “A logistic regression model based on inpatient health records to predict drug-induced liver injury caused by ramipril – an angiotensin-converting enzyme inhibitor” for publication in PLOS ONE. We appreciate the time and effort that you and the reviewers dedicated to providing feedback on our manuscript and are grateful for the insightful comments on and valuable improvements to our paper. We have incorporated most of the suggestions made by the reviewers. Please see below, editor/reviewer comments are black and our responses are in blue.

Editor comments

PLOS ONE's publication criteria require that the data presented in PLOS ONE manuscripts must support the conclusions drawn (http://journals.plos.org/plosone/s/criteria-for-publication#loc-4). We have some concerns about the conclusions in the manuscript. This is a retrospective observational study. However, here there are claims related to causation, e.g. "Our study demonstrated that using ramipril increases the risk of DILI" and "We found that taking ramipril increased the 2.68-fold risk of DILI (OR=2.68, p<0.001) compared to other ACEIs". Please revise your manuscript to address any instances of claims that are not supported by your results, e.g lines 251-252, lines 259-260 and lines 266-268 and also address the limitations of your study. 

Thank you for your comment. We revised our manuscript based on your suggestion.

Our study demonstrated that using ramipril increases the risk of DILI → From our study we have evidence that the odds of developing DILI is around two and a half times higher for ramipril using group compared non-use of ramipril

On the day of DILI, the ALT and ALP levels in patients who took ramipril were elevated by 8.2-and 14.1-fold, respectively → Taking ramipril group was observed to be higher risk of developing DILI compared to other ACEIs

The limitation was revised as below:

This case-control study tested a hypothesis about the correlation between intake of ramipril and DILI. Admittedly, our study is not as powerful as the other studies with a different study design in terms of establishing a causal relationship. Nevertheless, our study provided data and a reference for future clinical studies that would be using more rigorous scientific methods. Another limitation of our study is the patient's adherence to medication because our study subjects are outpatients. However, to measure adherence to drug therapy, we used the patient’s self report in every hospital visit. Moreover, identification of the cause of DILI may be influenced by certain drugs or supplements that cause DILI, but to the best of our knowledge, this has not been reported in the literature.

---

## [Decision Letter · Decision Letter 2]

23 May 2022

PONE-D-21-13763R2A logistic regression model based on inpatient health records to predict drug-induced liver injury caused by ramipril – an angiotensin-converting enzyme inhibitorPLOS ONE

Dear Dr. Nguyen,

Thank you for submitting your manuscript to PLOS ONE. After careful consideration, we feel that it has merit but does not fully meet PLOS ONE’s publication criteria as it currently stands. Therefore, we invite you to submit a revised version of the manuscript that addresses the points raised during the review process.

We look forward to receiving your revised manuscript.

Kind regards,

Jorge Enrique Machado-Alba, M.D; Ph.D

Academic Editor

PLOS ONE

Reviewers' comments:

Reviewer's Responses to Questions

**Comments to the Author**

1. If the authors have adequately addressed your comments raised in a previous round of review and you feel that this manuscript is now acceptable for publication, you may indicate that here to bypass the “Comments to the Author” section, enter your conflict of interest statement in the “Confidential to Editor” section, and submit your "Accept" recommendation.

Reviewer #3: (No Response)

2. Is the manuscript technically sound, and do the data support the conclusions?

Reviewer #3: No

3. Has the statistical analysis been performed appropriately and rigorously? 

Reviewer #3: No

4. Have the authors made all data underlying the findings in their manuscript fully available?

Reviewer #3: No

5. Is the manuscript presented in an intelligible fashion and written in standard English?

Reviewer #3: No

6. Review Comments to the Author

Reviewer #3: This study is about establishing approaches to predict the ramipril induced liver injury. Authors selected 1686 inpatients to generate 3 models predicting DILI by using logistic regression. But the processing of building risk estimation models is not clear, what are the criteria to keep those variables in model 1, 2, 3, respectively. Authors stated they used 2-fold cross-validation technique to evaluate the predictive performance, but the validation evaluation results with a1 and a2 were not presented. When doing the performance measurement by using ROC curve, authors did not provide the risk scores and corresponding cutoff points of each model. Based on the ROC curves, three models have perfect performances of predicting IDLI, but it was too good to be true because all three models can nearly predict every DILI case with very small part of false positive. If this paper just only wanted to identify the risk factors for ramipril induced liver injure prevention, the odd ratio (OR) of multiple regression is enough for drawing reliable conclusions.

Furthermore, Table 2 is not necessary because it is not relevant to the study purposes. Instead, characteristics of included patients (1686) should be presented in results part.

7. PLOS authors have the option to publish the peer review history of their article (what does this mean?). If published, this will include your full peer review and any attached files.

Reviewer #3: **Yes: **Yong Zhang

---

## [Author Response · Author response to Decision Letter 2]

26 May 2022

Dear editor and reviewers,

Thank you for giving us the opportunity to submit a revised draft of the manuscript “A logistic regression model based on inpatient health records to predict drug-induced liver injury caused by ramipril – an angiotensin-converting enzyme inhibitor” for publication in PLOS ONE. We appreciate the time and effort that you and the reviewers dedicated to providing feedback on our manuscript and are grateful for the insightful comments on and valuable improvements to our paper. We have incorporated most of the suggestions made by the reviewers. Please see below, editor/reviewer comments are black and our responses are in blue.

Reviewer comments

This study is about establishing approaches to predict the ramipril induced liver injury. Authors selected 1686 inpatients to generate 3 models predicting DILI by using logistic regression. But the processing of building risk estimation models is not clear, what are the criteria to keep those variables in model 1, 2, 3, respectively. Authors stated they used 2-fold cross-validation technique to evaluate the predictive performance, but the validation evaluation results with a1 and a2 were not presented. When doing the performance measurement by using ROC curve, authors did not provide the risk scores and corresponding cut off points of each model. Based on the ROC curves, three models have perfect performances of predicting IDLI, but it was too good to be true because all three models can nearly predict every DILI case with very small part of false positive. If this paper just only wanted to identify the risk factors for ramipril induced liver injure prevention, the odd ratio (OR) of multiple regression is enough for drawing reliable conclusions. 

Furthermore, Table 2 is not necessary because it is not relevant to the study purposes. Instead, characteristics of included patients (1686) should be presented in results part.

Q1: This study is about establishing approaches to predict the ramipril induced liver injury. Authors selected 1686 inpatients to generate 3 models predicting DILI by using logistic regression. But the processing of building risk estimation models is not clear, what are the criteria to keep those variables in model 1, 2, 3, respectively.

Response:

Thank you for your comments. We have revised the Methods section of our manuscript as follows:

The logistic regression model was built using the R statistical software, version 3.2.4 (A Language and Environment for Statistical Computing, Vienna, Austria) (14). To develop the DILI risk estimation model, we mapped the variables collected by the MHR to 23 discrete variables. The criteria for the inclusion of a variable in our model were based on clinical and statistical significance. We used forward selection based on the chi-square test of the change in residual deviance. Our goal was to build a logistic regression model that minimizes the variables until the optimal model that describes the data is found. We constructed three risk estimation models. These models used the presence or absence of DILI as the dependent variable and the 23 aforementioned variables as independent variables. We used a cutoff value of P<0.001 to add new terms. Differences among groups were analyzed using the chi-square test for qualitative variables, one-way analysis of variance for continuous variables with normal distributions, and the non-parametric Kruskal–Wallis test for continuous variables with non-standard distributions. 

Q2: Authors stated they used 2-fold cross-validation technique to evaluate the predictive performance, but the validation evaluation results with a1 and a2 were not presented.

Response:

Thank you for your comment.

We created multiple random splits of the dataset and divided them into training and validation datasets. For each split, the model was fitted to the training data, and predictive accuracy was assessed using the two-cross validation method. The results were then averaged over the splits.

The model performance is presented in Table 4 including specificity, sensitivity, and AROC

Q3: When doing the performance measurement by using ROC curve, authors did not provide the risk scores and corresponding cutoff points of each model.

Response:

Thank you for your suggestion. 

The Youden index method was used to find the optimal cut-off point, which is the difference between the number of true positives and the number of false positives over all possible cut-off point values. I would like to add the cut-off point in each model in Table 4 as follows:

Model Cutoff point Specificity Sensitivity Accuracy P-value# AROC! 95% CI$

Model 1 0.024 96.08 94.34 95.97 0.993 0.987 – 0.998

Model 2 0.026 96.58 98.11 96.68 0.37 0.995 0.991 – 0.999

Model 3 0.023 96.46 92.45 95.64 0.002 0.992 0.986 – 0.999

#: Anova test compared to model 1

AROC: Area under the receiver operating characteristic curve; CI: confidence interval

Q4: Based on the ROC curves, three models have perfect performances of predicting IDLI, but it was too good to be true because all three models can nearly predict every DILI case with very small part of false positive. If this paper just only wanted to identify the risk factors for ramipril induced liver injure prevention, the odd ratio (OR) of multiple regression is enough for drawing reliable conclusions.

Response:

Thank you for your valuable comments. We strongly agree that OR is sufficient to explore the risk factors for ramipril-induced DILI. However, we also wanted to build a model that includes all significant variables to predict the risk of DILI caused by ramipril, supporting decision-making in clinical practice.

 Q5: Furthermore, Table 2 is not necessary because it is not relevant to the study purposes. Instead, characteristics of included patients (1686) should be presented in results part.

Response:

Thank you for comments. We would like to add the characteristics of all patients (1686 patients) to Table 1. We would like to retain table 2 because it supports the selection of variables in the model.

---

## [Decision Letter · Decision Letter 3]

31 May 2022

PONE-D-21-13763R3

A logistic regression model based on inpatient health records to predict drug-induced liver injury caused by ramipril – an angiotensin-converting enzyme inhibitor

PLOS ONE

Dear author,

Thank you for submitting your manuscript to PLOS ONE. After careful consideration, we feel that it has merit but does not fully meet PLOS ONE’s publication criteria as it currently stands. Therefore, we invite you to submit a revised version of the manuscript that addresses the points raised during the review process.

We look forward to receiving your revised manuscript.

Kind regards,

Jorge Enrique Machado-Alba, M.D; Ph.D

Academic Editor

PLOS ONE

Additional Editor Comments (if provided):

Dear authors

We send you the comments of the evaluators of your manuscript so that you can carry out the corrections.

Reviewers' comments:

Reviewer's Responses to Questions

**Comments to the Author**

1. If the authors have adequately addressed your comments raised in a previous round of review and you feel that this manuscript is now acceptable for publication, you may indicate that here to bypass the “Comments to the Author” section, enter your conflict of interest statement in the “Confidential to Editor” section, and submit your "Accept" recommendation.

Reviewer #3: (No Response)

2. Is the manuscript technically sound, and do the data support the conclusions?

Reviewer #3: No

3. Has the statistical analysis been performed appropriately and rigorously? 

Reviewer #3: No

4. Have the authors made all data underlying the findings in their manuscript fully available?

Reviewer #3: No

5. Is the manuscript presented in an intelligible fashion and written in standard English?

Reviewer #3: No

6. Review Comments to the Author

Reviewer #3: The revised paper added more details and solved some of my concerns. But still I have little confidence about the reliability of those three models, because they are too good to be true (I can’t believe it!). Authors, based on their models, created 3 new indexes. We can call these indexes as risk score and authors this time provided the optimal cutoff points of those 3 risk scores for predicting of DILI. By using these risk scores, we then could perfectly predict the DILI. It is out of common sense because the components of those risk scores have weak or not strong impacts on DILI with margin positive OR values (in table 4). Therefore, in order to justify the performance of those models, author should double check their analysis and provide the details of those 3 risk scores in their studying population.

7. PLOS authors have the option to publish the peer review history of their article (what does this mean?). If published, this will include your full peer review and any attached files.

Reviewer #3: **Yes: **Yong Zhang

---

## [Author Response · Author response to Decision Letter 3]

2 Jun 2022

Dear editor and reviewers,

Thank you for giving us the opportunity to submit a revised draft of the manuscript “A logistic regression model based on inpatient health records to predict drug-induced liver injury caused by ramipril – an angiotensin-converting enzyme inhibitor” for publication in PLOS ONE. We appreciate the time and effort that you and the reviewers dedicated to providing feedback on our manuscript and are grateful for the insightful comments on and valuable improvements to our paper. We have incorporated most of the suggestions made by the reviewers. Please see below, editor/reviewer comments are black and our responses are in blue.

Editor comments

Reviewer #3: The revised paper added more details and solved some of my concerns. But still I have little confidence about the reliability of those three models, because they are too good to be true (I can’t believe it!). Authors, based on their models, created 3 new indexes. We can call these indexes as risk score and authors this time provided the optimal cutoff points of those 3 risk scores for predicting of DILI. By using these risk scores, we then could perfectly predict the DILI. It is out of common sense because the components of those risk scores have weak or not strong impacts on DILI with margin positive OR values (in table 4). Therefore, in order to justify the performance of those models, author should double check their analysis and provide the details of those 3 risk scores in their studying population.

We thank you for your thoughtful suggestions and insights. The manuscript has benefited from these insightful suggestions. 

All steps of data analysis, model construction and validation were rechecked by our researchers. We admit that there was an error in the AROC calculation due to using the wrong data set in R software. We have checked and corrected it according to your suggestion. Also, we have added the Fig 4 to better describe the details of our model.

Table 5. Comparison of the performances and the area under the ROC curves of the models.

Model Cutoff point Specificity Sensitivity P-value# AROC! 95% CI$ AIC R2

Model 1 0.029 0.884 0.727 0.831 0.766-0.897 147.17 0.375

Model 2 0.024 0.865 0.727 0.32 0.825 0.758-0.892 143.69 0.366

Model 3 0.033 0.896 0.682 0.29 0.820 0.752-0.888 142.8 0.372

Figure 4. Comparison of receiver operating characteristic curve curves of 3 models. AUC: area under the receiver operating characteristic curve

The limitation of our study was revised as below:

This case-control study tested a hypothesis about the correlation between intake of ramipril and DILI. Admittedly, our study is not as powerful as the other studies with a different study design in terms of establishing a causal relationship. Nevertheless, our study provided data and a reference for future clinical studies that would be using more rigorous scientific methods. Another limitation of our study is the patient's adherence to medication because our study subjects are outpatients. However, to measure adherence to drug therapy, we used the patient’s self-report in every hospital visit. Moreover, identification of the cause of DILI may be influenced by certain drugs or supplements that cause DILI, but to the best of our knowledge, this has not been reported in the literature. Although the independent variables were statistically significant related to DILI appreance our model performance had a low R-squared value. The R2 value is an overall measure of strength of association between the model and the response and does not reflect the contribution of independent predictor variable.

---

## [Decision Letter · Decision Letter 4]

29 Jun 2022

PONE-D-21-13763R4

A logistic regression model based on inpatient health records to predict drug-induced liver injury caused by ramipril – an angiotensin-converting enzyme inhibitor

PLOS ONE

Dear Dr. Phuong Thu Nguyen, MD, PhD,

Thank you for submitting your manuscript to PLOS ONE. After careful consideration, we feel that it has merit but does not fully meet PLOS ONE’s publication criteria as it currently stands. Therefore, we invite you to submit a revised version of the manuscript that addresses the points raised during the review process.

We await your responses to complete the process with your manuscript.

Please submit your revised manuscript on August 13, 2022. If you will need more time than this to complete your revisions, please reply to this message or contact the journal office at plosone@plos.org. Please include the following items when submitting your revised manuscript:

We look forward to receiving your revised manuscript.

Kind regards,

Jorge Enrique Machado-Alba, M.D; Ph.D

Academic Editor

PLOS ONE

Journal Requirements:

Reviewers' comments:

Reviewer's Responses to Questions

**Comments to the Author**

1. If the authors have adequately addressed your comments raised in a previous round of review and you feel that this manuscript is now acceptable for publication, you may indicate that here to bypass the “Comments to the Author” section, enter your conflict of interest statement in the “Confidential to Editor” section, and submit your "Accept" recommendation.

Reviewer #3: All comments have been addressed

2. Is the manuscript technically sound, and do the data support the conclusions?

Reviewer #3: Yes

3. Has the statistical analysis been performed appropriately and rigorously? 

Reviewer #3: Yes

4. Have the authors made all data underlying the findings in their manuscript fully available?

Reviewer #3: No

5. Is the manuscript presented in an intelligible fashion and written in standard English?

Reviewer #3: Yes

6. Review Comments to the Author

Reviewer #3: New version of the paper is much more reasonable. In methods part, there is a content of model evaluation with 2-fold cross-validation (page 7), but the evaluation results were absent. I believe readers like me may have some interests about the results of model evaluation.

7. PLOS authors have the option to publish the peer review history of their article (what does this mean?). If published, this will include your full peer review and any attached files.

Reviewer #3: **Yes: **Yong Zhang

---

## [Author Response · Author response to Decision Letter 4]

30 Jun 2022

Dear editor and reviewers,

Thank you for giving us the opportunity to submit a revised draft of the manuscript “A logistic regression model based on inpatient health records to predict drug-induced liver injury caused by ramipril – an angiotensin-converting enzyme inhibitor” for publication in PLOS ONE. We appreciate the time and effort that you and the reviewers dedicated to providing feedback on our manuscript and are grateful for the insightful comments on and valuable improvements to our paper. We have incorporated most of the suggestions made by the reviewers. Please see below, editor/reviewer comments are black and our responses are in blue.

Reviewers' comments:

6. Review Comments to the Author

Reviewer #3: New version of the paper is much more reasonable. In methods part, there is a content of model evaluation with 2-fold cross-validation (page 7), but the evaluation results were absent. I believe readers like me may have some interests about the results of model evaluation.

We thank you for your thoughtful suggestions and insights. The manuscript has benefited from these insightful suggestions. 

To validate the model, we used the ‘rms’ package on R software. The results of the 2-cross validation process are shown in Table 5 and Figure 4. In which we calculated the specificity, Sensitivity, AROC, AIC, R2 of 3 models.

Table 5. Comparison of the performances and the area under the ROC curves of the models.

Model Cut-off point Specificity Sensitivity P-value# AROC! 95% CI$ AIC R2

Model 1 0.029 0.884 0.727 0.831 0.766-0.897 147.17 0.375

Model 2 0.024 0.865 0.727 0.32 0.825 0.758-0.892 143.69 0.366

Model 3 0.033 0.896 0.682 0.29 0.820 0.752-0.888 142.8 0.372

AROC: Area under the receiver operating characteristic curve; AIC: Akaike information criterion

Figure 4. Comparison of receiver operating characteristic curve curves of 3 models. AUC: area under the receiver operating characteristic curve

Model 1 validation are as follows:

M1=lrm(DILI ~ age + sex + bmi + di_as + liver + total_dose_24 + bilt1 + gots +gpt1,data=default_tst)

pred.logit1=predict(M1)

phat1=1/(1+exp(-pred.logit1))

val.prob(phat1,default_trn$DILI,m=20,cex=.5, mkh=.02,)

head(predict(model_glm1, type = "response"))

model_glm_pred1 = ifelse(predict(model_glm1, type = "response") >0.029 , "1", "0")

print(model_glm_pred1)

length(default_tst$DILI)

length(model_glm_pred1)

length(default_trn$DILI)

calc_class_err = function(actual, predicted) {

 mean(actual != predicted)}

calc_class_err(actual = default_tst$DILI, predicted = model_glm_pred1)

train_tab1 = table(predicted = model_glm_pred1, actual = default_tst$DILI)

train_con_mat1 = confusionMatrix(train_tab1,positive="1")

c(train_con_mat1$overall["Accuracy"], 

 train_con_mat1$byClass["Sensitivity"], 

 train_con_mat1$byClass["Specificity"])

multi_class_rates <- function(confusion_matrix) {

 true_positives <- diag(confusion_matrix)

 false_positives <- colSums(confusion_matrix) - true_positives

 false_negatives <- rowSums(confusion_matrix) - true_positives

 true_negatives <- sum(confusion_matrix) - true_positives -

 false_positives - false_negatives

 return(data.frame(true_positives, false_positives, true_negatives,

 false_negatives, row.names = names(true_positives)))}

---

## [Decision Letter · Decision Letter 5]

12 Jul 2022

PONE-D-21-13763R5

A logistic regression model based on inpatient health records to predict drug-induced liver injury caused by ramipril – an angiotensin-converting enzyme inhibitor

PLOS ONE

Dear Dr. Nguyen,

Thank you for submitting your manuscript to PLOS ONE. After careful consideration, we feel that it has merit but does not fully meet PLOS ONE’s publication criteria as it currently stands. Therefore, we invite you to submit a revised version of the manuscript that addresses the points raised during the review process.

We look forward to receiving your revised manuscript.

Kind regards,

Jorge Enrique Machado-Alba, M.D; Ph.D

Academic Editor

PLOS ONE

Journal Requirements:

Reviewers' comments:

Reviewer's Responses to Questions

**Comments to the Author**

1. If the authors have adequately addressed your comments raised in a previous round of review and you feel that this manuscript is now acceptable for publication, you may indicate that here to bypass the “Comments to the Author” section, enter your conflict of interest statement in the “Confidential to Editor” section, and submit your "Accept" recommendation.

Reviewer #3: (No Response)

2. Is the manuscript technically sound, and do the data support the conclusions?

Reviewer #3: Partly

3. Has the statistical analysis been performed appropriately and rigorously? 

Reviewer #3: No

4. Have the authors made all data underlying the findings in their manuscript fully available?

Reviewer #3: No

5. Is the manuscript presented in an intelligible fashion and written in standard English?

Reviewer #3: Yes

6. Review Comments to the Author

Reviewer #3: Authors responded me with some R software codes, but not really addressed my concerns.

In this study, whole dataset was divided into two subsets to do cross-validation. Each subset will generate an optimal model with logistical regression method. Therefore, there should be two models for validation by the other subset. Then, by comparing MSE (mean squared error), Then the best model will be decided (validated). Last time, I asked for author to present the validation results. Authors responded that table 5 and figure 4 were the validation results. But I think those are results of performance evaluation.

Usually, we use one dataset or subset to generate one model, i.e. one regression equation. But the 3 datasets or subsets for each 3 model were not clearly given in this paper. Therefore, methods about the 3 models should be described with more specific details.

Besides, in page 7 line 127, about model development, “differences among groups were analyzed…”, What does the group in this sentence refer to? and results about groups differences were also not provided.

7. PLOS authors have the option to publish the peer review history of their article (what does this mean?). If published, this will include your full peer review and any attached files.

Reviewer #3: No

---

## [Author Response · Author response to Decision Letter 5]

14 Jul 2022

Dear editor and reviewers,

Thank you for giving us the opportunity to submit a revised draft of the manuscript “A logistic regression model based on inpatient health records to predict drug-induced liver injury caused by ramipril – an angiotensin-converting enzyme inhibitor” for publication in PLOS ONE. We appreciate the time and effort that you and the reviewers dedicated to providing feedback on our manuscript and are grateful for the insightful comments on and valuable improvements to our paper. We have incorporated most of the suggestions made by the reviewers. Please see below, editor/reviewer comments are black and our responses are in blue.

Reviewers' comments:

Reviewer #3: Authors responded me with some R software codes, but not really addressed my concerns.

In this study, whole dataset was divided into two subsets to do cross-validation. Each subset will generate an optimal model with logistical regression method. Therefore, there should be two models for validation by the other subset. Then, by comparing MSE (mean squared error), Then the best model will be decided (validated). Last time, I asked for author to present the validation results. Authors responded that table 5 and figure 4 were the validation results. But I think those are results of performance evaluation.

Usually, we use one dataset or subset to generate one model, i.e. one regression equation. But the 3 datasets or subsets for each 3 model were not clearly given in this paper. Therefore, methods about the 3 models should be described with more specific details.

Besides, in page 7 line 127, about model development, “differences among groups were analyzed…”, What does the group in this sentence refer to? and results about groups differences were also not provided.

We thank you for your thoughtful suggestions and insights. The manuscript has benefited from these insightful suggestions. 

For models that predict a continous variable, mean squared error is an ideal performance benchmark because of its link to the concept of cross-entropy from information theory. However, our outcome variable is binary with the estimation of the DILI risk with ramipril. 

Brier score was used to check the goodness of a predicted probability score. Brier score is very similar to the mean squared error, but only applied for prediction probability scores, whose values range between 0 and 1

In medical research, the Brier score (BS) and the area under the receiver operating characteristic ROC curve are two common metrics used to evaluate prediction models of a binary outcome, such as using biomarkers to predict the risk of developing a disease in the future.

Therefore, we would like to revise our method part as below:

Performance Measures

We measured the performance of the three models using the outcome (probability of DILI) of the test set obtained by 2-fold cross-validation. We plotted and measured the area under the receiver operating characteristics (ROC) curve of Model-1, Model-2, and Model-3 using the probability of DILI for comparing three areas under ROC curves obtained from the same dataset (15). Brier score was used to check the goodness of a predicted probability score (16). Lower Brier scores exhibited improved precision (17). The Youden index method was used to determine the optimal cut-off point, which is the difference between the number of true positives and the number of false positives over all possible cut-off points.

Additionally, we added the Brier score in Table 5 that would make the performance mesurement of the models becoming more clearly.

Table 5. Comparison of the performances and the area under the ROC curves of the models.

Model Cut-off point Specificity Sensitivity P-value# AROC! Brier scrore 95% CI$ AIC R2

Model 1 0.029 0.884 0.727 0.824 0.039 0.766-0.897 147.17 0.362

Model 2 

(- sex, BILT) 0.024 0.865 0.727 0.32 0.83 0.04 0.758-0.892 143.69 0.370

Model 3 

(-age, sex, BILT) 0.033 0.896 0.682 0.29 0.827 0.028 0.752-0.888 142.8 0.366

We found that model 3 was better with the smallest Brier score and showed that AROC was not significantly different from model 1 and model 2 (p>0.05).

---

## [Decision Letter · Decision Letter 6]

27 Jul 2022

A logistic regression model based on inpatient health records to predict drug-induced liver injury caused by ramipril – an angiotensin-converting enzyme inhibitor

PONE-D-21-13763R6

Dear Dr. Phuong Thu Nguyen,

We’re pleased to inform you that your manuscript has been judged scientifically suitable for publication and will be formally accepted for publication once it meets all outstanding technical requirements.

Kind regards,

Jorge Enrique Machado-Alba, M.D; Ph.D

Academic Editor

PLOS ONE

Additional Editor Comments (optional):

Dear authors

Congratulations, your work has been accepted for publication.

Jorge Machado

Academic Editor

---

## [Editor Report · Acceptance letter]

8 Aug 2022

PONE-D-21-13763R6 

A logistic regression model based on inpatient health records to predict drug-induced liver injury caused by ramipril – an angiotensin-converting enzyme inhibitor 

Dear Dr. Thi Thu:

I'm pleased to inform you that your manuscript has been deemed suitable for publication in PLOS ONE. Congratulations! Your manuscript is now with our production department. 

Kind regards, 

on behalf of

Dr. Jorge Enrique Machado-Alba 

Academic Editor

PLOS ONE